# Copy Number Variation of the *PIGY* Gene in Sheep and Its Association Analysis with Growth Traits

**DOI:** 10.3390/ani10040688

**Published:** 2020-04-15

**Authors:** Ziting Feng, Xinyu Li, Jie Cheng, Rui Jiang, Ruolan Huang, Dingchuan Wang, Yongzhen Huang, Li Pi, Linyong Hu, Hong Chen

**Affiliations:** 1Key Laboratory of Animal Genetics, Breeding and Reproduction of Shaanxi Province, College of Animal Science and Technology, Northwest A&F University, Yangling 712100, China; FengZT0629@163.com (Z.F.); lixinyu1502@126.com (X.L.); chengjie9412@163.com (J.C.); jiangrui613102@163.com (R.J.); huangruolan9904@163.com (R.H.); wdc135ccc@163.com (D.W.); hyzsci@126.com (Y.H.); 2Key Laboratory of Adaptation and Evolution of Plateau Biota, Northwest Institute of Plateau Biology, Chinese Academy of Sciences, Xining 810001, China; pili@nwipb.cas.cn (L.P.); xiangchou812@163.com (L.H.)

**Keywords:** copy number variation (CNV), *PIGY* gene, growth traits, sheep

## Abstract

**Simple Summary:**

The *PIGY* (phosphatidylinositol glycan anchor biosynthesis class Y) gene is a member of the PIG gene family and encodes the glycosylphosphatidylinositol-N-acetylglucosaminyltransferase (GPI-GnT) complex. It initiates the biosynthesis of GPI and plays an important role in cell–cell interactions. Sequencing has revealed a 3600 bp copy number variation (CNV) in exon 2 of the *PIGY* gene in sheep, potentially altering a functional part of the protein. The CNV overlaps 28 quantitative trait loci that are relevant to some economic traits like muscle density and carcass weight. We screened for this CNV of the *PIGY* gene in 569 individuals, namely, 240 Chaka sheep (CKS), 168 Hu sheep (HS), and 161 small-tailed Han sheep (STHS), and analyzed the association between the presence of this CNV and sheep body size traits. The results showed that the loss-type CNV was more prevalent than other types in these three breeds, and there were significant effects of the *PIGY* gene CNV on body weight, chest circumference, and circumference of cannon bone of sheep. The results showed that sheep with gain-type CNV had better growth traits than those with other types. The findings reveal the relationship between the CNV of the *PIGY* gene and growth traits of sheep, suggesting that CNV could be utilized for improved molecular breeding of sheep.

**Abstract:**

Copy number variation (CNV) is a type of genomic variation with an important effect on animal phenotype. We found that the *PIGY* gene contains a 3600 bp copy number variation (CNV) region located in chromosome 6 of sheep (Oar_v4.0 36,121,601–36,125,200 bp). This region overlaps with multiple quantitative trait loci related to phenotypes like muscle density and carcass weight. Therefore, in this study, the copy number variation of the *PIGY* gene was screened in three Chinese sheep breeds, namely, Chaka sheep (CKS, May of 2018, Wulan County, Qinghai Province, China), Hu sheep (HS, May of 2015, Mengjin County, Henan Province, China), and small-tailed Han sheep (STHS, May of 2016, Yongjing, Gansu Province, China). Association analyses were performed on the presence of CNV and sheep body size traits. We used real-time quantitative PCR (qPCR) to detect the CNV for association analysis. According to the results, the loss-type CNV was more common than other types in the three breeds (global average: loss = 61.5%, normal = 17.5%, and gain = 21.0%). The association analysis also showed significant effects of the *PIGY* gene CNV on body weight, chest circumference, and circumference of the cannon bone of sheep. Sheep with gain-type CNV had better growth traits than those with other types. The results indicate a clear relationship between the *PIGY* gene CNV and growth traits of sheep, suggesting the use of CNV as a new molecular breeding marker.

## 1. Introduction

Copy number variation (CNV) is a kind of variation ranging from 50 kb to several Mb in size compared with the normal genome sequence of organisms [1]. Its forms include duplication, deletion, insertion, translocation, and derived chromosomal structural variation. CNV is an important genetic basis for individual differences. This type of variation is widely distributed in the human genome. It covers more nucleotide polymorphisms than single nucleotide polymorphisms (SNPs) and greatly enriches the diversity of genetic variation in the genome [2]. In addition, copy number variation has been demonstrated to contribute to animal phenotypic polymorphism and disease susceptibility, making it an important genetic variation mode. CNVs related to important traits have been screened in humans and many animal models, providing a theoretical basis for molecular studies of human disease and animal breeding [3,4]. Quantitative trait loci (QTL) includes regions of DNA that are associated with a particular phenotypic trait, which varies in degree and can be attributed to polygenic effects [5]. Growth-related traits are generally considered to be quantitative traits, which may be controlled by single genes or several major genes. The addition of environmental effects can make the traits exhibit continuous variation. Overall, QTL is an important reference index to evaluate relationships between genes and growth traits.

*PIGY* (phosphatidylinositol glycan anchor biosynthesis class Y) gene is a member of the PIG gene family, which is located on chromosome 6 of sheep. Its expression product, together with the expression proteins of *PIGA, PIGC, PIGH, PIGP, PIGQ*, and *DPM2* genes, forms the glycosylphosphatidylinositol-N-acetylglucosaminyltransferase (GPI-GnT) complex, which initiates the biosynthesis of GPI [6]. GPI, an anchor for many surface proteins, is synthesized on the endoplasmic reticulum. Until now, studies on the *PIGY* gene have mainly focused on human diseases, such as deformities, epilepsy, severe stunting, cataracts, and early death. The study of the *PIGY* gene in sheep is still in the early stage. However, sequencing analysis has revealed that the CNV of the *PIGY* gene overlaps with 28 QTL that are related to sheep economic traits like muscle density and carcass weight. Thus, the *PIGY* may be a key gene influencing important economic traits of sheep.

Sheep is an important species of domestic animals. They are timid, gentle, and easy to tame, making them popular for breeding worldwide. Sheep provide products such as meat and fur for humans, with meat that is tender and tasty. In 2015, the world’s sheep stock totaled 1.16 billion, with 158 million in China. The world output of sheep meat is 9.261 million tons, making it the fourth largest meat species [7]. Thus, the way people can improve the economic characteristics of sheep meat has become a focus of research in the field of animal husbandry. In recent years, scientists have made some progress in the field of sheep CNV research. In 2010, Fontanesi et al. [8] designed a smooth array containing 385,000 oligonucleotide probes using the bovine genome and preliminarily obtained a comparative map of CNVs in the sheep genome. In 2013, Liu et al. analyzed 329 individuals of three sheep breeds (Sunit, Dubois, and German meat sheep) using SNP50 chips for sheep and detected 238 CNV regions, including 73 CNV regions with frequencies greater than 3% [9]. In 2016, Zhu et al. [10] found that genes such as *PPARA*, *RXRA*, and *KLF11* were related to fat deposition and affected the tail type of sheep. These CNV regions contain a large number of genes associated with fat metabolism and GTPase activity. In 2017, Ma et al. [11] used SNP microarray to study 48 Chinese Tan sheep and determined 1296 CNV regions, accounting for 4.7% of the sheep genome.

Based on the above findings, we hypothesized that the identified CNV of the *PIGY* gene is an important genetic marker for economic traits of sheep. The goal of this work was to explore the effect of copy number variation of the *PIGY* gene in three Chinese sheep breeds, namely, Chaka sheep (CKS), Hu sheep (HS), and small-tailed Han sheep (STHS), and examine the association between the presence of this CNV and economic traits of these three sheep breeds. The results of this work should provide the theoretical basis for future molecular breeding of sheep in China.

## 2. Materials and Methods

### 2.1. Animal and Growth Trait Measurements

The timeline of the experimental process is shown in Figure 1.

We selected a total of 569 individuals from three sheep breeds to study changes in the copy number of the *PIGY* gene, including CKS (*n* = 240, May 2018, Wulan County, Qinghai Province, China), HS (*n* = 168, May 2015, Mengjin County, Henan Province, China), and STHS (*n* = 161, May 2016, Yongjing, Gansu Province, China). CKS, HS, and STHS are common sheep breeds in China. CKS is a semifine wool sheep used for both wool and meat in the Qinghai plateau. This breed is native to the area around the Chaka Salt Lake in Wulan County. Influenced by the unique geographical climate and soil environment, the mutton from this sheep breed is rich in minerals and vitamins, and the meat quality is tender and fat but not greasy. Hu sheep exhibit excellent characteristics, including strong fecundity, suitable feeding, rapid early growth, and good meat performance and quality, making this breed important in China’s sheep industry. STHS is a kind of sheep cultivated in China for both meat and fur production. This breed has characteristics of fast growth, strong reproduction, and strong adaptability. It is praised as the “national treasure” of China, “super sheep”, and “high-legged sheep” of the world. To summarize, the sheep varieties selected for this experiment have important value and status in China’s sheep industry, and analysis of variations linked to improved characteristics of these breeds would benefit China’s livestock production.

We measured the different economic traits of the sheep breeds. For CKS, we measured age, gender, body weight, body height, body length, and chest circumference. For HS, we measured age, gender, body weight, body length, chest circumference, tube circumference, body height, and rump width. For STHS, we measured age, gender, chest deep, tube circumference, height at hip cross, chest width, chest circumference, body height, and body length. The minimum value, mean value (sd) and maximum value of each trait are shown in Appendix A.

In this study, all animal testing was conducted in accordance with applicable international animal guidelines, animal testing welfare acts, and related policies. All experiments met the requirements of this study and followed the guidance of animal welfare law and related policies of the International Faculty Animal Policy and Welfare Committee of Northwest A&F University (FAPWC-NWAFU, protocol number, NWAFAC1008).

### 2.2. Sample Collection and Genomic DNA Isolation

Genomic DNA was extracted from the blood of all experimental sheep by phenol–chloroform extraction [12], and the concentration of DNA samples was measured by Nanodrop 2000 spectrophotometer. Then, the DNA samples were diluted to 25 ng/μL and stored at −40 °C for subsequent use. Finally, no sheep suffered any injuries or health problems during blood collection in this study. 

### 2.3. Identification of Target Gene and Internal Reference Gene and Primer Design

Using the *PIGY* gene sequence of sheep published in the NCBI database as the reference sequence, we found a 3600 bp copy number variation (Oar_v4.0 36,121,601–36,125,200 bp) region in exon 2 of the *PIGY* gene in sheep (Figure 2) (Huang et al., unpublished data), potentially altering a functional part of the protein. This CNV overlapped 28 quantitative trait loci related to sheep economic traits, such as muscle density and carcass weight (Figure 3). The *ANKRD1* gene appeared stable with two copies (Huang et al., unpublished data), and according to the Animal Omics Database (http://animal.nwsuaf.edu.cn/), the *ANKRD1* gene is present in two copies in both cattle and sheep. Therefore, we selected the *ANKRD1* gene as a diploid internal reference. The information of two PCR primer pairs are listed in Table 1, which were designed by Primer-BLAST from the NCBI website. The size of the amplified fragment in the candidate region of the *PIGY* gene was 90 bp, and the size for amplification of the *ANKRD1* gene was 143 bp. 

### 2.4. Testing of Primers

We used real-Time quantitative polymerase chain reaction (qPCR) to determine whether the primers were suitable by drawing the amplification curve and the dissolution peak. As shown in Figure 4, the curves were consistent for the samples. The curve trend was smooth, the curve peak was high and sharp, and no primer dimers or nonspecific amplification products were observed.

### 2.5. Copy Number Variation Detection of the PIGY Gene

We used qPCR to determine the copy number of potential CNVs in 569 sheep. The amplification system was performed in 10.0 μL reactions containing the following: 25 ng/μL template DNA (genomic DNA extracted from blood samples), 1 μL; 10 pmol/μL upstream and downstream primers, 0.5 μL; 2 × SYBR Green qPCR Mix, 5 μL; ddH_2_O, 3 μL. Each sample was assayed in triplicate reactions [13]. Thermal cycling conditions were as follows: 95 °C for 1 min followed by 39 cycles of 95 °C for 15 s, 60 °C for 15 s, and 72 °C for 30 s. The solution curve started at 65 °C, increased 0.5 °C for each cycle, reached 95 °C, and was then measured for 5 s. Melting curve analysis was completed at the end of the amplification with the following conditions: starting from 65 °C, each cycle increased by 0.5 °C for 0.05 s and increased to 95 °C. Finally, we determined the average Ct value of each sample to calculate the copy number of each sheep DNA sample [14].

### 2.6. Statistical Analyses of Data

We used the calculation formula 2 × 2^−∆Ct^ [15,16,17] to determine the copy number of each sheep sample. Three types of copy number (gain, loss, and normal) were classified as >2, <2, and 2 copies, respectively. Then, we used SPSS v22.0 software (SPSS, Inc., Chicago, IL, USA) to analyze the association between the copy number and economic traits of each sheep, with age and sex considered as fixed factors so as to calculate the distribution of different copy number variation types of the *PIGY* gene and the dominant variation types of each trait. Individuals of each breed were unrelated and from the same farm, and the phenotypes were measured in the same season. 

In the analysis of genotype effect, a fixed model was adopted: *Y_ijk_* = *μ* + *A_i_* + *S_j_* + *CNV_j_* + *e_ijk_*(1)
where *Y_ijk_* is the observed value of traits, *μ* is the mean value of the population, *A_i_* is the age of each individual, *S_j_* is the effect of sex, *CNV_j_* is the fixed effect of the *PIGY* gene’s copy number variation type, and *e_ijk_* is the random error [18,19]. Each breed was analyzed separately. LSD multiple comparison was used to test the differences between each group of data, and the results were expressed as mean values ± SE. Finally, the variance between varieties was tested by chi-square. 

## 3. Results

### 3.1. Distribution of Copy Number Variation Types in Three Sheep Breeds

Based on the qPCR results, we calculated the *PIGY* gene copy number of 569 sheep by 2 × 2^−∆Ct^. The sequencing results revealed a CNV region of the *PIGY* gene (Chr6: 36,121,601–36,125,200 bp), and the presence of CNV was analyzed for three sheep breeds, as shown in Figure 5A. The data revealed the presence of CNV in this region of the genome. The copy number for each sheep DNA sample was classified as >2, <2, and 2 copies for gain, loss, and normal, respectively. As shown in Figure 5B, the frequencies of copy number polymorphisms in the three sheep breeds revealed that loss-type CNV was the most frequent in all three breeds. The specific frequencies are shown in Table 2. Chi-squared values showed a significantly different CNV distribution among the three sheep breeds (Table 3), indicating that the *PIGY* CNV may be breed-specific.

### 3.2. Association Analyses among Copy Number Variation and Sheep Growth Traits

Previous studies have shown that copy number variation of genes can affect the growth traits of animals, so breeding should be tailored to the dominant variation types of animals. In this study, we used the general linear model to associate the economic traits of three sheep breeds to identify the dominant variation type of the *PIGY* gene in sheep. Association analysis revealed that the CNV of the *PIGY* gene had a significant effect on body weight and chest circumference in CKS and an extremely significant effect on the circumference of the cannon bone in STHS. However, there was no significant difference between CNV and growth traits of HS. Sheep with gain-type CNV exhibited significantly higher body weight, chest circumference, and circumference of cannon bone than those with loss- or normal-type CNV in CKS and STHS, with a significance of *p* < 0.01 (Table 4).

## 4. Discussion

The results support our hypothesis that the CNV of the *PIGY* gene is an important genetic marker of economic traits of sheep. Copy number variation (CNV) can be used as a marker in molecular breeding to select important economic traits of livestock and facilitate disease diagnosis [20]. Many copy number variations have potential effects on the economic traits of domestic animals [21]. In addition to CNV, SNPs are another important source of phenotypic variation in animals, and SNP loci are also used in sheep breeding as molecular genetic markers. SNP analysis studies have uncovered important genetic effects, but the identified SNPs explain only a small proportion of the phenotypic variation, so the predictive power of these SNPs remains low for many complex traits [22]. CNV may be an alternative marker [23]. Compared with SNP, CNVs are longer fragments of variation with more significant effects on the growth traits of sheep. More and more copy number variation studies have been performed on different species, such as human [24], mouse [25], cattle [26,27], chicken [28], monkey [29], and sheep [30,31]. In recent years, with increasing consumer requirements for sheep quality and the development of molecular breeding, there is a need to exploit CNV molecular genetic variation for economic trait selection and disease prevention. With the development of gene chip technology and new generation sequencing technology, copy number variation will be more efficiently and more widely used in future sheep genetic breeding, creating more development possibilities for the sheep industry in China.

In recent years, few studies have explored the *PIGY* gene, and most works have focused on human diseases as this gene encodes the anchoring protein GPI complex. This is the first report of the *PIGY* gene in sheep. We found that the *PIGY* gene in sheep contains a copy number variation region located in exon 2, and it overlaps with 28 QTLs, including carcass weight and muscle density. Previous studies have reported that, in chickens, some QTLs associated with CNV fragments are related to body weight, carcass weight, and pectoral muscle mass [32,33]. Therefore, we focused on the *PIGY* gene to investigate the association between copy number variation and sheep economic traits.

In this study, we detected CNV of the *PIGY* gene in three breeds and its association with growth traits and found that the three breeds showed the same distribution of CNV, with the deletion of copy number variations as the main type. Beyond that, there were significant relationships between multiple growth traits and copy number in Chaka sheep and small-tailed Han sheep, such as body weight, chest circumference, and circumference of cannon bone, and individuals with gain-type CNV had an obvious advantage in these growth traits. However, no significant relationship between the *PIGY* gene CNV and growth traits was observed in Hu sheep. According to the chi-squared values (χ^2^) in CNV among breeds (Table 3), the copy number variation of the *PIGY* gene had significant difference among breeds. We speculate that this result is due to differences in the genetic background and living environment of these different sheep breeds, which will lead to positional uncertainty of CNV on the genome. Chaka sheep and small-tailed Han sheep are from the plateau region of China, but Hu sheep are native to the flatlands. The difference in environments may contribute to the difference among the three breeds, but we cannot eliminate a potential link between this result and the difference in phenotype among the three breeds. These aspects may explain why a significant relationship between CNV and growth traits was not detected in all three varieties. Chaka sheep and small-tailed Han sheep with increased copy number exhibited better phenotypes than other types. Growth traits like body weight, chest circumference, and circumference of cannon bone are all closely related to the economic value of sheep. The data suggest that *PIGY* gene copy number variation could be used as a molecular genetic marker of sheep, allowing improvement to the quality of sheep and an accelerated breeding process. Sheep products play an increasingly important role in the Chinese market, and molecular breeding can be an effective strategy to improve the quality of sheep. This work links the *PIGY* gene CNV to sheep growth characteristics, providing a new molecular marker for sheep breeding. Determination of functional differences of copy number variation among different breeds and details of the mechanism of action require further experimental research.

## 5. Conclusions

This is the first analysis of the *PIGY* CNVs in three Chinese sheep breeds. The CNV of the *PIGY* gene overlaps with 28 quantitative trait loci that are related to sheep economic traits, suggesting a promising impact of the *PIGY* gene on growth traits of sheep, so we hypothesize that the CNV of the *PIGY* gene is an important genetic marker for economic traits. Additional association analysis supports this hypothesis. Our study provides preliminary evidence for a functional role of the *PIGY* CNV in larger populations and different sheep breeds, which might provide new insights into the potential applications of CNVs as new promising molecular markers in animal breeding.

## Figures and Tables

**Figure 1 animals-10-00688-f001:**
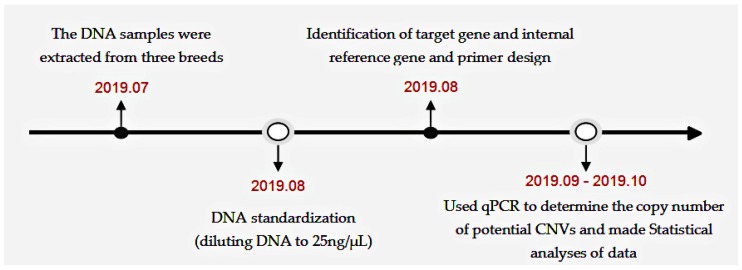
The overall flow diagram of the experiment.

**Figure 2 animals-10-00688-f002:**
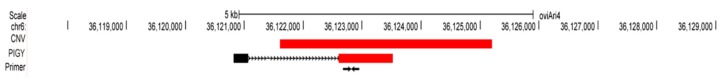
The region of the *PIGY* gene’s copy number variation (CNV) in sheep breeds. The CNV region: Chr6: Oar_v4.0 36,121,601–36,125,200 bp (Huang et al., unpublished data); the *PIGY* gene region: Chr6: Oar_v4.0 36,120,820–36,123,522 bp; pp (primer pair CNV): 36,122,803–36,122,892 bp. The detection sequence size is 90 bp.

**Figure 3 animals-10-00688-f003:**
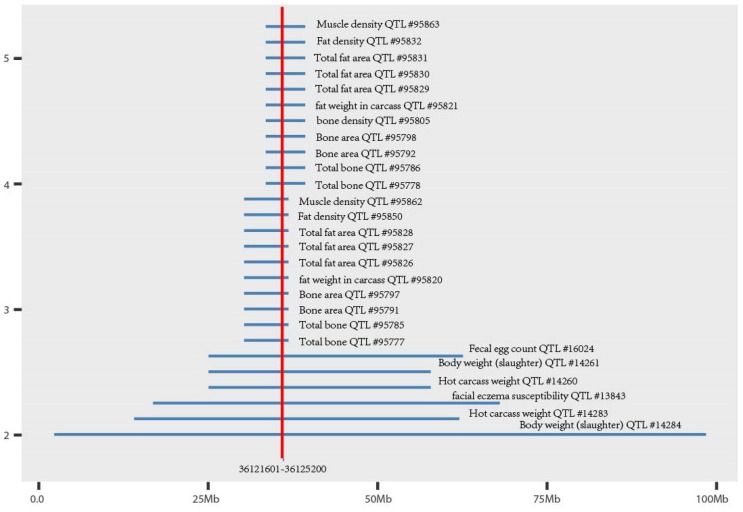
The CNV region of the *PIGY* gene overlaps with the quantitative trait loci (QTL) of the sheep.

**Figure 4 animals-10-00688-f004:**
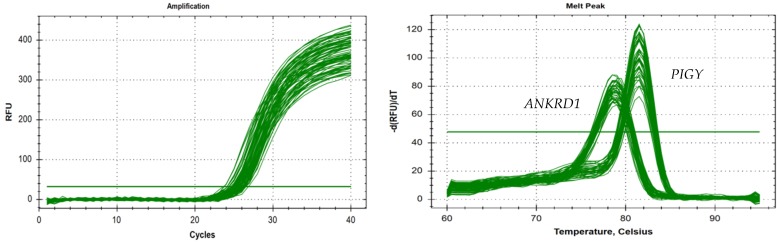
The specificity testing of the *PIGY* gene and the *ANKRD1* gene.

**Figure 5 animals-10-00688-f005:**
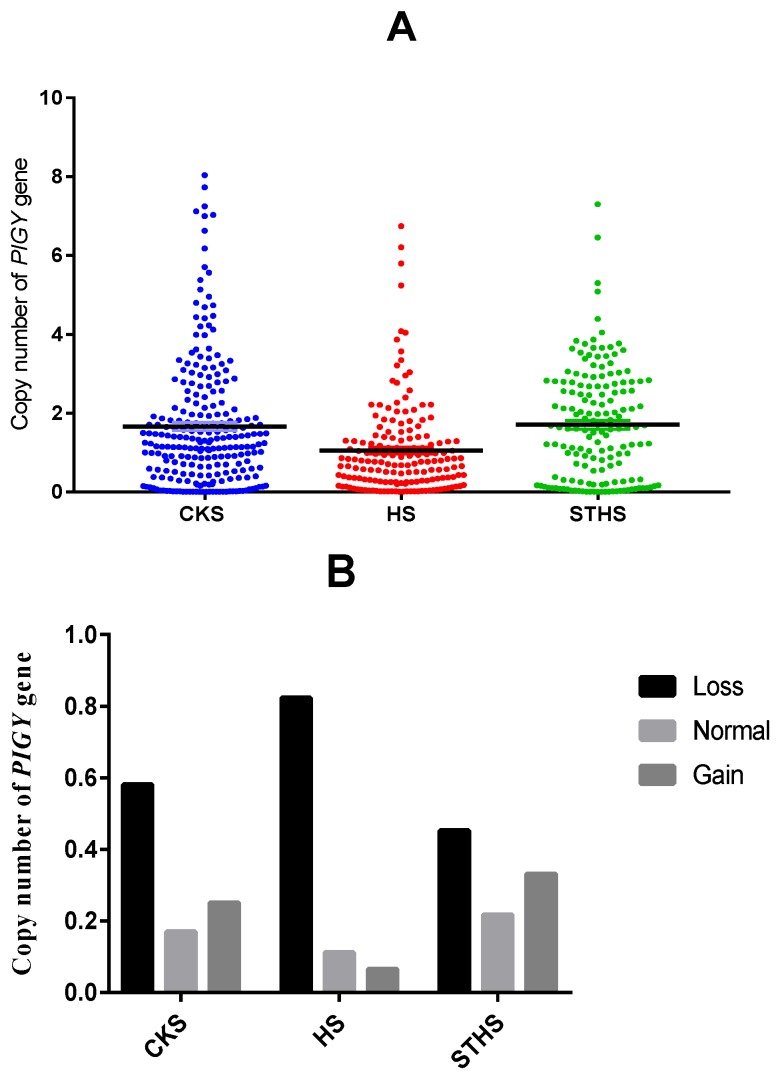
(**A**) Distribution of the *PIGY* gene’s CNV in three sheep breeds. (**B**) Frequency of the copy numbers of the *PIGY* gene’s CNV in three sheep breeds. CK: Chaka sheep, HS: Hu sheep, STHS: small-tailed Han sheep.

**Table 1 animals-10-00688-t001:** The primer information used in this study.

Primer Pairs	Name	Genes	Sequences	Amplification Length (Bp)	Tm (°C)
*PIGY*-CNV	F1	*PIGY*	5′-AGAGTGGCGGGTGATAAGTG-3′	90 nt	59.46
	R1		5′-CAGTCCTGCCAAAGACACCA-3	90 nt	60.18
*ANKRD1*-CNV	F2	*ANKRD1*	5′-TGGGCACCACGAAATTCTCA-3′	143 nt	60.00
	R2		5′-TGGCAGAAATGTGCGAACG-3′	143 nt	60.00

F: forward primer; R: reverse primer.

**Table 2 animals-10-00688-t002:** Frequencies of different CNV types in three sheep breeds.

Breeds	Loss (%)	Normal (%)	Gain (%)
CKS	0.604	0.171	0.225
HS	0.780	0.131	0.089
STHS	0.460	0.224	0.316

CKS: loss (*n* = 145), normal (*n* = 41), gain (*n* = 54). HS: loss (*n* = 131), normal (*n* = 22), gain (*n* = 15). STHS: loss (*n* = 74), normal (*n* = 36), gain (*n* = 51).

**Table 3 animals-10-00688-t003:** The chi-square test of CNV types among Chinese sheep breeds.

Breeds	CKS	HS	STHS
CKS		16.285 (*p* = 2.91 × 10^−4^)	8.183 (*p* = 7 × 10^−3^)
HS			38.733 (*p* = 3.88 × 10^−9^)
STHS			

Chi-squared values (χ^2^) for differences in CNV between two breeds.

**Table 4 animals-10-00688-t004:** Statistical association analysis of ovine *PIGY* gene CNV with growth traits of three Chinese sheep breeds.

Breeds	Growth Traits	CNV Type (Mean ± SE)			*p*
		Loss (*n* = 350)	Normal (*n* = 99)	Gain (*n* = 120)	
CKS	Body weight (kg)	58.801 ± 3.084 ^a^	54.280 ± 3.156 ^b^	61.065 ± 3.353 ^aC^	0.027 *
	Chest circumference (cm)	92.544 ± 2.036 ^A^	89.076 ± 2.084 ^B^	93.251 ± 2.214 ^AC^	0.029 *
STHS	Circumference of cannon bone (cm)	6.872 ± 0.082 ^A^	7.298 ± 0.116 ^B^	7.342 ± 0.095 ^BC^	0.000 **

^a,b^ Value that differ significantly at *p* < 0.05; ^A,B,C^ value that differ significantly at *p* < 0.01; * value that differ significantly at *p* < 0.05; ** value that differ significantly at *p* < 0.01. CKS: loss (*n* = 145), normal (*n* = 41), gain (*n* = 54). HS: loss (*n* = 131), normal (*n* = 22), gain (*n* = 15). STHS: loss (*n* = 74), normal (*n* = 36), gain (*n* = 51).

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
