# Peer review of "Copy Number Variation of the PIGY Gene in Sheep and Its Association Analysis with Growth Traits"

_animals, 2020, doi:10.3390/ani10040688_

Round 1
Reviewer 1 Report
ANIMALS; Referee’s Evaluation Report
MANUSCRIPT IDENTIFICATION:
ANIMALS-735101 REVISED VERSION-MARCH-26-2020
COPY NUMBER VARIATION OF PIGY GENE IN SHEEP AND ITS ASSOCIATION ANALYSIS WITH GROWTH TRAITS (RESEARCH ARTICLE)
Comments to Authors/Editor:
The paper of Feng and colleagues investigated the possible association among the copy number variation (CNV) of PIGY gene and different growth traits in Chaka (CKS), Hu (HS) and Small Tail Han (STHS) sheep in order to unveil new molecular marker approaches in animal breeding. This Revised Version of the paper of Feng was sent to Rhine Language Edition, Beijing, China for language-grammar-syntaxes edition. The manuscript has been greatly improved yet, there were still detected some minor grammar issues in such a revised version. The simple summary and the abstract section were corrected as suggested. In this reviewed version, the authors included the requested information regarding the sheep evaluated genotypes while incorporated evidence regarding the main quantitative outcomes obtained in the three breeds. In the introduction section, the authors added info with respect to the economic and social importance of the breeds chosen for the study; the working hypothesis was included. In the M&M section, the authors included a figure with the timeline of actions throughout the experimental process. Besides, important information regarding the social, environmental and productive importance of the sheep breeds was also added. The authors specified the main economic traits evaluated in the three different genotypes. The statistical analyses subsection was corrected, including the required new information. In the Results section, the authors also included new information to better understand the main outcomes of the quantitative and qualitative approaches. Some titles of Figures and Tables were rewritten, although not in a homogeneous fashion. In the Discussion section, the authors started the section stating the link between the working hypothesis with respect to the main results obtained. Some other suggested issues were also included in this Revised Version. The authors clarified the main association among the CVN-PIGY-gene with respect to the collected growth traits in the studied sheep breeds. They also mentioned the observed environment differences; this interesting interaction (breed, environment & genotype) may explain why a significant relationship among CNV and growth traits in all three genotype-variation within breed was not detected. Conclusions were rewritten; in this new version, the authors did not do not need to include a mini-summary…, congratulations!!! In this new version, the authors highlighted the main findings of the study were aligned with respect to both the objectives and the working hypothesis and proposed their findings as preliminary evidence for the interesting ole of the PIGY-CNV in larger populations and different sheep breeds mentioned these research outcomes as a new promising molecular marker not only limited to sheep breeding but with potential applications in the animal industry. As I mentioned, this interesting study deserved a better-written conclusion; they made it.
Author Response
Dear reviewer: Thank you very much for your valuable advice. It's our pleasure to have your approval of our work. According to your suggestion, we have checked and corrected the grammar issues in the whole manuscript. Also, we further examined and modified the format of the pictures and tables. The revised part is marked in blue in the manuscript.Reviewer 2 Report
The manuscript is improved greatly compared to its previous version; its more clear, comprehensible and readable than its previous version. However, I am of the opinion that the manuscript still require a moderate English language correction. Apart from that, I have two minor comments:
Material and methods:
1). Line 175-177: "to analyze the association between copy number and economic traits of each sheep, with age and sex considered as fixed factors..", so it appears from this sentence that the authors also took into account the sex of an individual as the fixed factor. However, in line 180 (where they actually show the model), they do not include the "sex" in the model. Why is this discrepancy?
Result:
In table-3, where they show the result of chi-square test to determine whether the proportion of CNV types differ among the Chinese sheep breeds. However, it is not clear that whether they used the total CNV counts,gained CNVs or loss CNVs in this calculation.
Author Response
Dear Reviewer:
Thanks for your comments on our manuscript entitled “Copy number variation of PIGY Gene in Sheep and Its Association” (ID: animals-769922). Those comments are all valuable and very helpful for revising and improving our paper. We have read the comments carefully and have made correction. Revised parts are marked in blue in the paper. The corrections are as followings:
1. Response to comment: The manuscript is improved greatly compared to its previous version; its more clear, comprehensible and readable than its previous version. However, I am of the opinion that the manuscript still require a moderate English language correction.
Response: Thank you very much for your valuable advice. It's our pleasure to have your approval of our work. According to your suggestion, we have checked and corrected the grammar issues in the whole manuscript. Revised parts are marked in blue in the paper.
2. Response to comment: Material and methods: Line 175-177: "to analyze the association between copy number and economic traits of each sheep, with age and sex considered as fixed factors..", so it appears from this sentence that the authors also took into account the sex of an individual as the fixed factor. However, in line 180 (where they actually show the model), they do not include the "sex" in the model. Why is this discrepancy?
Response: Thank you very much for your advice. We are very sorry for our negligence that we didn’t write the “sex” in the model. We have corrected the model in the manuscript.(L180, L182)
3. Response to comment: Result: In table-3, where they show the result of chi-square test to determine whether the proportion of CNV types differ among the Chinese sheep breeds. However, it is not clear that whether they used the total CNV counts,gained CNVs or loss CNVs in this calculation.
Response: Thank you very much for your advice. When we analyze the data, we use SPSS 22.0 to calculate the chi-square according to the data in the table below. In the “Breeds” column, “1” is Chaka sheep, “2” is Hu sheep, “3” is small-tailed Han sheep. In the “CNV Type” column, “1” is Loss type, “2” is Normal type, “3” is Gain type. Then, we used these data to carry out chi-square tests among three breeds.
| Breeds | CNV Type | Frequency |
| 1 | 1 | 145 |
| 1 | 2 | 41 |
| 1 | 3 | 54 |
| 2 | 1 | 131 |
| 2 | 2 | 22 |
| 2 | 3 | 15 |
| 3 | 1 | 74 |
| 3 | 2 | 36 |
| 3 | 3 | 51 |
This manuscript is a resubmission of an earlier submission. The following is a list of the peer review reports and author responses from that submission.
Round 1
Reviewer 1 Report
ANIMALS
Referee’s Evaluation Report
MANUSCRIPT IDENTIFICATION: ANIMALS-735101
Copy number variation of PIGY gene in sheep and its association analysis with growth traits (RESEARCH ARTICLE)
Comments to Authors/Editor:
The paper of Feng and colleagues investigated the possible association among the copy number variation (CNV) of PIGY gene and different growth traits in Chaka (CKS), Hu (HS) and Small Tail Han (STHS) sheep in order to unveil new molecular marker approaches in animal breeding. This manuscript falls within the scope of ANIMALS. The manuscript is sufficiently informative for the replication of the study. In general, the organization of the experiment seems to be well designed. Nonetheless, English quality, grammar, and sentence structure need to be reviewed. Both the simple summary and the abstract were written in a careless fashion; L20, “analysis between” or “analyses among”…? Besides, please avoid to include both the term “significant differences” along with the probability level itself; it is a pleonasm, correct accordingly along the whole manuscript sections. The sheep genotypes used in the experiment were declared. L37, … “was more”…, more what? If no differences between experimental groups were declared, please include the global average for the response variables. Please include in the abstract the location, and time of the year when the samples were taken. I was wondering if there is any possibility of a seasonal-effect upon the expression of, either the PIGY gene or their CNV´s evaluated??? In the Introduction section, English grammar and syntaxes are not standard. L84, …” on its growth traits??? While the objectives of the study were clearly stated, a working hypothesis was not declared; this is a must. In the Material & Methods section, it is recommended to include information regarding the prevailing conditions when the sampling process occurred. Besides, a figure including the main timeline of action regarding the experimental schedule would be really helpful for the readership to better understand the study. The genetic background of the animals used in the study was indicated, yet, it should be important to define why these genotypes Chaka, Hu, and Small Tailed Han) was used for the study; are they important in the area of study from an economic or social viewpoint? Please clarify. The ethical approval of the study from the home institution was included, yet the authors did not mention if the study was conducted according to the protocols of an international guide for the ethical use of animals in research; this is a must. Again, the authors must include when the blood samples were collected. Figures 1 & 2 are quite important, yet the resolution quality is poor; please enhance the quality. The materials, standards, and methods used are relevant and in accordance with the objectives of the study. Also, all the treatment, sampling techniques, molecular and genomic methods, as well as response variables considered in the experiment are detailed and accurate, while in accordance with the objectives of the study. L138, … “to detect the 569 sheep.”; a non-sense sentence or incomplete sentence; correct accordingly. The experimental design is described well enough for the reader to understand how the experiment was carried out. L149, 2.6. “Statistical analysis…” or “Statistical analyses…”???. English quality must be improved also in this section. Besides, as already mentioned and quite important, please improve the quality of the presented Figures; in their actual format, they are not clear enough; the definition quality is quite poor. When analyzing the response variables, where was included in the model the effect of the genotype upon the value traits??? In Figure 4, and Table 2, please confirm if there is a genotype effect upon the CNV; is there any difference regarding the CNV distribution and frequencies among breeds??; please confirm. Again, please avoid to include both the term “significant differences” along with the probability level itself; it is a pleonasm, correct accordingly along the Results section and some parts in the Discussion section. L177; between or among??? The novelty value of the results is reasonable, with different variables included in the study. The results were shown in tables and figures; titles must be rewritten. Certainly, the titles of both tables and figures must stand by themselves. Again, please improve the quality of the presented Figures; in their actual format, they are not clear enough. Besides, the authors must homogenize the format of tables; in their actual format, they are quite dissimilar. At the beginning of the Discussion section, I do strongly suggest initiating this section including the working hypothesis of the study. Authors must define if, with the obtained results, such a hypothesis is non-rejected or rejected. L206, …” we think…”, or …” based in our results, we propose that….”??? For this reason, the authors must include the working hypothesis prior to the objectives in the Introduction section. L204-205; different animals or different species??? L216; …” fresh but not full of mutton”???; what do the authors mean by that??? L231-232; … “were used to analyze.”; to analyze what???, this is a non-sense sentence; please correct accordingly. L234; …” What´s more…???!!!; poor grammar. As previously commented, is it possible to see an interaction sheep-age x genotype x month-season of blood sampling upon the PIGY gene expression of upon the CNV??? In general, the authors made an accurate interpretation of the main findings. Nonetheless, Conclusions must be rewritten; the authors do not need to include a mini-summary in this section. Please just highlight the main findings of your study and the possible use of the study outcomes upon sheep breeding; conclusions must be aligned with the working hypothesis. Please rewrite this section; this interesting study deserves a better-written conclusion. The list of references cited in the manuscript is proper. This is an interesting study, yet, as already stated, the English is not of sufficient standard. In fact, many sentences throughout the text are improperly phrased, and both incomplete and long sentences were detected along with the manuscript. Sorry about this situation but it is necessary to ensure that the paper is readable. The authors must increase the readability and the scientific writing and merit of the manuscript. At this point, and based on the above comments, my pronouncement is that this manuscript can be reconsidered after moderate revision.
Author Response
Dear Editors and Reviewe#1:
Thanks for your letter and the reviewers’ comments on our manuscript entitled “Copy number variation of PIGY Gene in Sheep and Its Association” (ID: animals-735106). Those comments are all valuable and very helpful for revising and improving our paper. We have read the comments carefully and have made correction. Revised parts are marked in yellow in the paper. The corrections are as followings:
1. Response to comment:The manuscript is sufficiently informative for the replication of the study. In general, the organization of the experiment seems to be well designed. Nonetheless, English quality, grammar, and sentence structure need to be reviewed. Both the simple summary and the abstract were written in a careless fashion
Response: Thank you very much for your valuable advice. We are very sorry for our incorrect grammar and sentence structure. According to your advice, we have our manuscript checked by Dr. Kevin Li, a native English speaking, and we further modified and improved the language of our manuscript, especially in the simple summary and the abstract. We temporarily put the certificate of language editing at the end of the manuscript.
2. Response to comment: L20, “analysis between” or “analyses among”…?
Response: Thank you for your advice. We are very sorry for our incorrect writing. We've rewritten the sentence in the article. (L20-21)
3. Response to comment: please avoid to include both the term “significant differences” along with the probability level itself; it is a pleonasm, correct accordingly along the whole manuscript sections
Response: Thank you for your valuable advice. We've avoided to include both the term “significant differences” along with the probability level itself. Corrections have been made in the paper. (L23-24、L40-41、L209-214)
4. Response to comment: L37, … “was more”…, more what?If no differences between experimental groups were declared, please include the global average for the response variables.
Response: Thank you for your valuable advice. We have included the global average for the response variables. Correction has been made in the paper.(L39)
5. Response to comment: Please include in the abstract the location, and time of the year when the samples were taken. I was wondering if there is any possibility of a seasonal-effect upon the expression of, either the PIGY gene or their CNV´s evaluated
Response: Thank you very much for your advice. According to your suggestion, we added the location and the time of the year when the samples were taken in the abstract. (L33-35) Actually, We took this into account that different sampling seasons may affect the phenotype, so in order to ruled out the seasonal-effect upon the expression in this experiment, all the phenotypes we used were measured in the same season. We are very sorry that we didn’t make it clear. We have explained it in the 2.6 Statistical analyses of data.(L178-179)
6. Response to comment: L84, …” on its growth traits??? While the objectives of the study were clearly stated, a working hypothesis was not declared;
Response: Thank you for your valuable advice. We corrected the statement error accordingly and we have added the working hypothesis in the manuscript.(L89-90).
7. Response to comment: In the Material & Methods section, it is recommended to include information regarding the prevailing conditions when the sampling process occurred. a figure including the main timeline of action regarding the experimental schedule would be really helpful for the readership to better understand the study.
Response: Thank you for your valuable advice. In order to give the reader a better understanding, we added the the overall flow diagram of experiment in the manuscript. (Figure 1)
8. Response to comment: The genetic background of the animals used in the study was indicated, yet, it should be important to definewhy these genotypes (Chaka, Hu, and Small Tailed Han) was used for the study;are they important in the area of study from an economic or social viewpoint? Please clarify.
Response: Thank you for your valuable advice. We are sorry that we don't really understand your meaning of this word ”genotypes”, so we respond to this question in two aspects:
(1) For the genotypes, we refer to the methods of relevant articles(Ref15-17) that normal copy number of sheep genome =2, so we used three types of genotypes (gain, loss and normal) were classified as >2, <2 and 2 copies and we mentioned it in the 2.6 Statistical analyses of data ( L174). Different genotypes (loss, normal, gain) will make the different growth status, So our goal in this study is to find the dominant genotype which can make more economic value.
(2) For the breeds we used in this research, according to your suggestion, we have added the utilization value of these three sheep (L104-113).
9. Response to comment: The ethical approval of the study from the home institution was included, yet the authors did not mention if the study was conducted according to the protocols of an international guide for the ethical use of animals in research;
Response: Thank you for your advice. We are sorry we didn't make it clear in the manuscript. All used animals were conducted in accordance with applicable international animal guidelines, and the Faculty Animal Policy and Welfare Committee of Northwest A&F University is according to international guide. We have made corrections in the manuscript.(L121, L124)
10. Response to comment: Again, the authors must include when the blood samples were collected.
Response: Thank you for your advice. Same as the Response 5, we have added the collection time and the overall flow diagram of experiment to the manuscript.(L33-35, Figure 1) Actually, We took this into account that different sampling seasons may affect the phenotype, so in order to ruled out the seasonal-effect upon the expression in this experiment, all the phenotypes we used were measured in the same season. We are very sorry that we didn’t make it clear. Correction has been made in the manuscript.(L178-179)
11. Figures 1 & 2 are quite important, yet the resolution quality is poor; please enhance the quality.
Response: Thank you for your advice. We have improved the quality of the pictures in the manuscript.(Figure 2, Figure 3)
12. Response to comment: L138, … “to detect the 569 sheep.”; a non-sense sentence or incomplete sentence; correct accordingly.
Response: Thank you for your advice. We are very sorry for our incorrect writing and the correction has been made in the manuscript.(L161)
13. Response to comment: L149, 2.6. “Statistical analysis…” or “Statistical analyses…”???. English quality must be improved also in this section.
Response: Thank you for your advice. We are very sorry for our incorrect writing. We have replaced “Statistical analysis” by “Statistical analyses” in the article, and we further modified and improved the language in this section. (L172)
14. Response to comment: As already mentioned and quite important, please improve the quality of the presented Figures; in their actual format, they are not clear enough; the definition quality is quite poor.
Response: Thank you for your advice. We have improved the quality of the pictures in the manuscript.(Figure 4)
15. Response to comment: When analyzing the response variables, where was included in the model the effect of the genotype upon the value traits
Response: Thank you very much for your good advice. We are sorry that we don't really understand your meaning of this word ”genotypes”, so we respond to this question in two aspects:
(1) For the effect of different genotype upon the value traits, in fact, in the fixed model : Yijk =μ+Ai +CNVj+ eijk, CNVj represent the genotype ( gain, normal, loss) and the effect of the genotype is a fixed effect(L180-183).
(2) For the effect of different breeds upon the value traits, when we used the model to analyze the data, we analyzed each breed separately, so the breed were consistent in every analysis. That is why we didn't put the breeds effect into the model. We are very sorry that we didn’t make it clear, and we have explained it in the manuscript(L183).
16. Response to comment: In Figure 4, and Table 2, please confirm if there is a genotype effect upon the CNV; is there any difference regarding the CNV distribution and frequencies among breeds
Response: Thank you for your valuable advice. Figures 4 have been adjusted to Figure 5 in the manuscript. We are sorry that we don't really understand your meaning of this word “genotypes”, so we respond to this question in two aspects:
(1) For the genotype effect upon the CNV, the Figure and tables were showed the distribution of different genotype (loss, normal, gain) , so there must be a CNV genotype effect. Different genotypes (loss, normal, gain) will make the different growth status(Table 4), and in our study, the Gain type was the dominant genotype of sheep(L212-214).
(2) For the breeds effect upon the CNV, we are very sorry that we didn't make it clear. We have added the chi-square values (χ2) in CNV among breeds in the manuscript.(Table 3) It can be seen from the analysis results that there is a significantly different CNV distribution among the three sheep breeds, indicating that different genetic backgrounds lead to significant differences in the distribution of PIGY CNV.
17. Response to comment: Again, please avoid to include both the term “significant differences” along with the probability level itself; it is a pleonasm, correct accordingly along the Results section and some parts in the Discussion section.
Response: Thank you for your valuable advice. Same as Response 3, we have made corrections in the manuscript.(L209-214)
18. Response to comment: L177; between or among???
Response: Thank you for your advice. We are very sorry for our incorrect writing. We have replaced “analysis between” by “analyses among” in the article. Same as Response 2, we have made corrections in the manuscript.(L205)
19. Response to comment: The results were shown in tables and figures; titles must be rewritten. Certainly, the titles of both tables and figures must stand by themselves
Response: Thank you for your advice. We are very sorry for our negligence. The title of figure 4 has been rewritten and the titles of both tables and figures have stood by themselves. Correction has been made in the paper.(L216)
20. Response to comment: Again, please improve the quality of the presented Figures; in their actual format, they are not clear enough.
Response: Thank you for your advice. We have improved the quality of the pictures in the manuscript.(Figure 5)
21. Response to comment: Besides, the authors must homogenize the format of tables; in their actual format, they are quite dissimilar.
Response: Thank you for your advice. We are very sorry for our negligence. We have made corrections in the manuscript to homogenize the format of tables.(Table1-4)
22. Response to comment: At the beginning of the Discussion section, I do strongly suggest initiating this section including the working hypothesis of the study. Authors must define if, with the obtained results, such a hypothesis is non-rejected or rejected.
Response: Thank you for your valuable advice. We've already initiating the Discussion including whether our working hypothesis is non-rejected or rejected. And based on our results, the initial working hypothesis is valid.(L221-222)
23. Response to comment: L206, …” we think…”, or …” based in our results, we propose that….”??? For this reason, the authors must include the working hypothesis prior to the objectives in the Introduction section.
Response: Thank you for your valuable advice. We are very sorry for our negligence. Same as Response 21, we have made corrections in the manuscript.(L221-222)
24. Response to comment: L204-205; different animals or different species???
Response: Thank you for your advice. We are very sorry for our incorrect writing. We have replaced “different animals” by “different species” in the manuscript.(L232)
25. Response to comment: L216; …” fresh but not full of mutton”???; what do the authors mean by that???
Response: We are very sorry for the incorrect expression caused by our negligence, this sentence has been deleted.
26. Response to comment: L231-232; … “were used to analyze.”; to analyze what???, this is a non-sense sentence; please correct accordingly.
Response: Thank you for your advice. We are very sorry for our incorrect writing. Correction has been made in the paper. We have deleted this sentence and rewitten this part(L248-259)
27. Response to comment: L234; …” What´s more…???!!!; poor grammar.
Response: Thank you for your advice. We have replaced “What´s more” by “Beyond that” in the manuscript.(L250)
28. Response to comment: As previously commented, is it possible to see an interaction sheep-age x genotype x month-season of blood sampling upon the PIGY gene expression of upon the CNV
Response: Thank you very much for your valuable advice. Based on Response 5 and Response 16, when we analyzed by SPSS v22.0 software, we had considered the interaction among sheep-age x genotype x month-season of blood sampling, but the results of SPSS analysis showed that the interaction among them was not significant, so we excluded this effect, and finally we decided to use this model: Yijk = μ+Ai +CNVj+ eijk, where Yijk is the observed value of traits;μis the mean value of population, Ai is the age of individual, CNVj was the fixed effect of PIGY gene’s copy number variation type, and eijk was the random error.(L180-183)
29. Response to comment: In general, the authors made an accurate interpretation of the main findings.Nonetheless, Conclusions must be rewritten; the authors do not need to include a mini-summary in this section. Please just highlight the main findings of your study and the possible use of the study outcomes upon sheep breeding; conclusions must be aligned with the working hypothesis.Please rewrite this section; this interesting study deserves a better-written conclusion.
Response: Thank you very much for your valuable advice. According to your suggestion, we have re-written the conclusions section, which included the main findings of our study and the possible use of the study outcomes upon sheep breeding. And the conclusions is aligned with our working hypothesis. Correction has been made in the paper. (L275-281)
30. Response to comment: This is an interesting study, yet, as already stated, the English is not of sufficient standard. In fact, many sentences throughout the text are improperly phrased, and both incomplete and long sentences were detected along with the manuscript. Sorry about this situation but it is necessary to ensure that the paper is readable. The authors must increase the readability and the scientific writing and merit of the manuscript.
Response: Thank you very much for your valuable advice. We are very sorry for our incorrect grammar and sentence structure. According to your advice, we have our manuscript checked by Dr. Kevin Li, a native English speaking, and we further modified and improved the language of our manuscript. We temporarily put the certificate of language editing at the end of the manuscript.
Reviewer 2 Report
The results presented are interesting but the discussion is a little unstructured. In addition, while the paper is generally well written, it does need editing for English to make it sparkle. Specific comments follow:
Lines 16-17 “... a 3600 bp copy number variation in the exon ...” should be “... a 3600 bp copy number variation (CNV)in the exon ...”
Line 20 It’s better to use “small-tailed Han sheep ” instead of “Small Tail Han sheep”
Line 20 “made” should be “implemented”
Lines 22-23,33,39 a space should be added before parentheses, please check carefully
Line 30 the parentheses is not entered in English,please check carefully.
Lines 55-53 Ref3. didn’t mention any human disease research
Line 55 “QTL” should be “quantitative trait loci (QTL)”
Line 64 A reference should be added after “which initiates the biosynthesis of glycosylphosphatidylinositol (GPI). “
Line 68 “quantitative tait loci” should be “QTL”.
Lines 74-76. Please provide the sheep meat production / total meat production in the world and China.
Lines 77-80 There are several published papers about sheep CNV:
[1] Liu J, Zhang L, Xu L, et al. Analysis of copy number variations in the sheep genome using 50K SNP BeadChip array[J]. BMC genomics, 2013, 14(1): 229.
[2] Zhu C, Fan H, Yuan Z, et al. Genome-wide detection of CNVs in Chinese indigenous sheep with different types of tails using ovine high-density 600K SNP arrays[J]. Scientific reports, 2016, 6: 27822.
[3] Yang L, Xu L, Zhou Y, et al. Diversity of copy number variation in a worldwide population of sheep[J]. Genomics, 2018, 110(3): 143-148.
[4]Jenkins G M, Goddard M E, Black M A, et al. Copy number variants in the sheep genome detected using multiple approaches[J]. BMC genomics, 2016, 17(1): 441.
[5] Ma Q, Liu X, Pan J, et al. Genome-wide detection of copy number variation in Chinese indigenous sheep using an ovine high-density 600 K SNP array[J]. Scientific reports, 2017, 7(1): 1-10.
[6]Fontanesi L, Beretti F, Martelli P L, et al. A first comparative map of copy number variations in the sheep genome[J]. Genomics, 2011, 97(3): 158-165.
Why didn’t mention these papers?
In “2.1 Animal and growth traits measurements” section,please explain why you use these three sheep breeds? I think lines 212-223 should be putted here instead of in discussion section.
Lines 94-98 please provide the min value, mean value (sd) and max value of these records.
Line 119 ANKRD1 should be italic
Line 156 Breed effects should be added in the fixed model.
Lines200-201. I’m not agree with that “Because of their own limitations, they has no significant effect on the regulation of gene expression ... ” , because many eQTL papers in farm animal has been published in recent years,e.g. Ponsuksili S, Murani E, Trakooljul N, et al. Discovery of candidate genes for muscle traits based on GWAS supported by eQTL-analysis[J]. International journal of biological sciences, 2014, 10(3): 327.
Line 205-206 There are several published papers about sheep CNV.
Author Response
Dear Editors and Reviewe#2:
Thanks for your letter and the reviewers’ comments on our manuscript entitled “Copy number variation of PIGY Gene in Sheep and Its Association” (ID: animals-735106). Those comments are all valuable and very helpful for revising and improving our paper. We have read the comments carefully and have made correction. Revised parts are marked in yellow in the paper. The corrections are as followings:
1. Response to comment: Lines 16-17 “... a 3600 bp copy number variation in the exon ...” should be “... a 3600 bp copy number variation (CNV)in the exon ...”
Response: Thank you very much for your suggestion. We have corrected accordingly in the manuscript.(L16)
2. Response to comment: Line 20 It’s better to use “small-tailed Han sheep ” instead of “Small Tail Han sheep”
Response: Thank you very much for your suggestion. We have corrected accordingly (L20) and checked it in the whole article.
3. Response to comment: Line 20 “made” should be “implemented”
Response: Thank you very much for your suggestion. Combined with the suggestions of Reviewer 1, we have rewritten this sentence in the manuscript.(L20-21)
4. Response to comment: Lines 22-23,33,39 a space should be added before parentheses, please check carefully
Response: Thank you very much for your suggestion.We are very sorry for our negligence. Combined with the suggestions of Reviewer 1, the contents in these brackets have been deleted in the manuscript. We have check and corrected other parentheses in the whole manuscript.
5. Response to comment: Line 30 the parentheses is not entered in English,please check carefully.
Response: Thank you very much for your suggestion. We have corrected accordingly in the manuscript.(L30)
6.Response to comment: Lines 55-53 Ref3. didn’t mention any human disease research
Response: Thank you very much for your suggestion. Here we have added references about the use of CNV in human diseases . Correction has been made in the manuscript.(L54-56, Ref4.)
7. Response to comment: Line 55 “QTL” should be “quantitative trait loci (QTL)”
Response: Thank you very much for your suggestion. We have corrected accordingly in the manuscript.(L56)
8. Response to comment: Line 64 A reference should be added after “which initiates the biosynthesis of glycosylphosphatidylinositol (GPI).
Response: Thank you very much for your suggestion. Here we have added references accordingly(L65, Ref6.)
9. Response to comment: Line 68 “quantitative tait loci” should be “QTL”
Response: Thank you very much for your suggestion. We have corrected accordingly in the manuscript.(L69)
10. Response to comment: Lines 74-76. Please provide the sheep meat production / total meat production in the world and China.
Response: Thank you very much for your suggestion. We have added the data about the sheep meat production in the world and China in the manuscript. (L75-77)
11.Response to comment: Lines 77-80 There are several published papers about sheep CNV:
[1] Liu J, Zhang L, Xu L, et al. Analysis of copy number variations in the sheep genome using 50K SNP BeadChip array[J]. BMC genomics, 2013, 14(1): 229.
[2] Zhu C, Fan H, Yuan Z, et al. Genome-wide detection of CNVs in Chinese indigenous sheep with different types of tails using ovine high-density 600K SNP arrays[J]. Scientific reports, 2016, 6: 27822.
[3] Yang L, Xu L, Zhou Y, et al. Diversity of copy number variation in a worldwide population of sheep[J]. Genomics, 2018, 110(3): 143-148.
[4]Jenkins G M, Goddard M E, Black M A, et al. Copy number variants in the sheep genome detected using multiple approaches[J]. BMC genomics, 2016, 17(1): 441.
[5] Ma Q, Liu X, Pan J, et al. Genome-wide detection of copy number variation in Chinese indigenous sheep using an ovine high-density 600 K SNP array[J]. Scientific reports, 2017, 7(1): 1-10.
- Fontanesi L, Beretti F, Martelli P L, et al. A first comparative map of copy number variations in the sheep genome[J]. Genomics, 2011, 97(3): 158-165.
Why didn’t mention these papers?
Response: Thank you very much for your valuable suggestion. We have read these papers carefully and added them in the manuscript. They are listed in Ref 9, Ref 10, Ref 30, Ref 31, Ref 11, Ref 8.
12. Response to comment: In “2.1 Animal and growth traits measurements” section,please explain why you use these three sheep breeds? I think lines 212-223 should be putted here instead of in discussion section.
Response:Thank you very much for your valuable suggestion. We are quite in favor of your proposal. We have moved this paragraph to“2.1 Animal and growth traits measurements” section.(L104-113)
13.Response to comment: Lines 94-98 please provide the min value, mean value (sd) and max value of these records.
Response: Thank you very much for your valuable suggestion. We put these in Table S1 as supplementary data.(L282-283)
14. Response to comment: Line 119 ANKRD1 should be italic
Response: We are very sorry for our negligence. We have corrected accordingly in the manuscript.(L143)
15. Response to comment: Line 156 Breed effects should be added in the fixed model.
Response:Thank you very much for your valuable suggestion. We are very sorry that we didn’t make it clear. In fact, when we used the model to analyze the data, we analyzed each breed separately, so the breed were consistent in every analysis. That is why we didn't put the breed effect into the model. We have explained it in the manuscript.(L183)
16.Response to comment: Lines200-201. I’m not agree with that “Because of their own limitations, they has no significant effect on the regulation of gene expression ... ” , because many eQTL papers in farm animal has been published in recent years,e.g. Ponsuksili S, Murani E, Trakooljul N, et al. Discovery of candidate genes for muscle traits based on GWAS supported by eQTL-analysis[J]. International journal of biological sciences, 2014, 10(3): 327.
Response: We are very sorry for the Incorrect expression caused by our negligence, we have corrected this sentence in the manuscript and added the corresponding references.(L227-229, Ref 22)
17. Response to comment: Line 205-206 There are several published papers about sheep CNV.
Response: We are very sorry for the Incorrect expression. we have corrected this sentence in the manuscript. (L233-235)
Reviewer 3 Report
Major comments:
1). How did the authors conclude that there is a 3.6 Kb long CNV encompassing PIGY gene in sheep genome? They mentioned several times that it is based on the sequencing results but they never show it or mentioned reference about it. Interestingly, there is no structural variations encompassing this PIGY gene in DGVa of Ensembl (http://www.ensembl.org/Ovis_aries/Gene/Summary?g=ENSOARG00000000447;r=6:36193017-36193229;t=ENSOART00000000475)
2). The authors never discuss about the character of this CNV; whether it encompass the entire gene or some specific exon. They should discuss it.
3). The association between various growth traits and CNV was reported only in Chaka sheep and Small Tailed Han Sheep, while the same could not be observed for Hu sheep. The authors do not give a satisfactory explanation for it. If CNV itself has an effect on the expression of underlying genes. Then why it is not observed in Hu sheep. Moreover, how did they rule out the effect of population structure in the association analysis?
Minor comments:
line 28:
Copy number variation is one types of animal genome variations: This sentence reads awkward because CNV is not exclusively found in animal genome only.
Introduction
line 67: “is still in the blank stage”: consider replacing this with “is still in early stage.”
line 72: “sheep are”: consider replacing this with “sheep is one of the important species of domestic animals”
Material and methods
1). Mention the version of sheep reference used in the paper.
2). Please provide the reference for this sentence: “chose ANKRD1 gene as the internal reference gene...”
3). In the legend of the figure 1: remove NC_019463.2 and just keep “chr6”. The legend is confusing in its present form.
Author Response
Dear Editors and Reviewe#3:
Thanks for your letter and the reviewers’ comments on our manuscript entitled “Copy number variation of PIGY Gene in Sheep and Its Association” (ID: animals-735106). Those comments are all valuable and very helpful for revising and improving our paper. We have read the comments carefully and have made correction. Revised parts are marked in yellow in the paper. The corrections are as followings:
1. Response to comment: How did the authors conclude that there is a 3.6 Kb long CNV encompassing PIGY gene in sheep genome? They mentioned several times that it is based on the sequencing results but they never show it or mentioned reference about it. Interestingly, there is no structural variations encompassing this PIGY gene in DGVa of Ensembl (http://www.ensembl.org/Ovis_aries/Gene/Summary?g=ENSOARG00000000447;r=6:36193017-36193229;t=ENSOART00000000475)
Response: Thank you very much for your valuable suggestion. We are very sorry that we didn't make it clear. The sequencing data in our study came from Huang et al. Since the study of Huang has not been published, we did not elaborate in detail in our manuscript, but the study showed that the QTL of PIGY gene was located in a region related to growth traits, and there is a research indicating that mutations in PIGY gene have been found in the human genome. (Ref 6: Ilkovski, B.; Pagnamenta, A.T.; O'Grady, G.L. et al. Mutations in PIGY: expanding the phenotype of inherited glycosylphosphatidylinositol deficiencies. Human molecular genetics. 2015, 24(21), 6146-6159. ) Thus, these are our reference for selecting PIGY gene for research. In order to make it easier for readers to understand, we explain it in the manuscript.( L135, L138, L146)
2. Response to comment: The authors never discuss about the character of this CNV; whether it encompass the entire gene or some specific exon. They should discuss it.
Response: Thank you very much for your valuable suggestion. According to the sequencing results, this CNV encompass exon 2 of the PIGY gene in sheep. We have added the character of this CNV in the Simple Summary, Materials and Methods and Discussion.(L16-17, L134-136, L241-244 )
3. Response to comment: The association between various growth traits and CNV was reported only in Chaka sheep and Small Tailed Han Sheep, while the same could not be observed for Hu sheep. The authors do not give a satisfactory explanation for it. If CNV itself has an effect on the expression of underlying genes. Then why it is not observed in Hu sheep. Moreover, how did they rule out the effect of population structure in the association analysis?
Response: Thank you very much for your valuable suggestion. We are very sorry that we didn’t make a clear explanation that why this results didn’t observed in Hu sheep. We speculate that this is due to differences in their genetic background and living environment. Although the gene CNV is present in the three breeds, the three sheep differentiate in the genetic background and living conditions, which will lead to positional uncertainty of CNV on the genome. In other words, the variation fragment may be inserted or deleted into different parts of the genome. In order to explain it clear, we added the chi-square values (χ2) in CNV among breeds in the manuscript.(Table 3) It can be seen from the analysis results that a significantly different CNV distribution among the three sheep breeds exist. The above may be can explain why the Hu sheep did not have the same result as Chaka and small tailed han sheep. We have explained it in the manuscript.(L255-263)
Besides, The reason that we rule out the effect of population structure in the association analysis is that in this study, each species was analyzed separately, so there was no effect of population structure in the association analysis. We are very sorry that we didn’t make it clear. We've explained it in the manuscript.(L185)
4. Response to comment: line 28: Copy number variation is one types of animal genome variations: This sentence reads awkward because CNV is not exclusively found in animal genome only.
Response: Thank you very much for your suggestion. We are very sorry for the Incorrect expression and we have corrected this sentence in the manuscript.(L28-29)
5. Response to comment: line 67: “is still in the blank stage”: consider replacing this with “is still in early stage.”
Response: Thank you very much for your suggestion. We have corrected accordingly in the manuscript.(L68)
6. Response to comment: Mention the version of sheep reference used in the paper.
Response: Thank you very much for your suggestion. In this study, we used Oar_v4.0 as the version of sheep reference. We have added it in the manuscript.(L134)
7. Response to comment: Please provide the reference for this sentence: “chose ANKRD1 gene as the internal reference gene...”
Response: Thank you very much for your suggestion. We selecting ANKRD1 gene as the internal reference gene is according to the animal omics database, where we found it is present in two copies in both cattle and sheep, and the ANKRD1 gene appeared stable with two copies (Huang et al., unpublished data). We have added links of this websites in the manuscript.(L138-139)
8. Response to comment: In the legend of the figure 1: remove NC_019463.2 and just keep “chr6”. The legend is confusing in its present form.
Response: Thank you very much for your suggestion. We have removed NC_019463.2 in the manuscript. Considering the explicit region of CNV, here we added the version of sheep reference we used (“Oar_v4.0”). (L146)